# Exercise induces new cardiomyocyte generation in the adult mammalian heart

Ana Vujic [1], Carolin Lerchenmüller [2,3], Ting-Di Wu[4,5], Christelle Guillermier[3,6,7], Charles P. Rabolli[2], Emilia Gonzalez[1], Samuel E. Senyo[8], Xiaojun Liu[2,3], Jean-Luc Guerquin-Kern[4,5], Matthew L. Steinhauser[3,6,7,9], Richard T. Lee[1] & Anthony Rosenzweig[2,3]

Loss of cardiomyocytes is a major cause of heart failure, and while the adult heart has a limited capacity for cardiomyogenesis, little is known about what regulates this ability or whether it can be effectively harnessed. Here we show that 8 weeks of running exercise increase birth of new cardiomyocytes in adult mice (~4.6-fold). New cardiomyocytes are identified based on incorporation of $^{15}N$-thymidine by multi-isotope imaging mass spectrometry (MIMS) and on being mononucleate/diploid. Furthermore, we demonstrate that exercise after myocardial infarction induces a robust cardiomyogenic response in an extended border zone of the infarcted area. Inhibition of miR-222, a microRNA increased by exercise in both animal models and humans, completely blocks the cardiomyogenic exercise response. These findings demonstrate that cardiomyogenesis can be activated by exercise in the normal and injured adult mouse heart and suggest that stimulation of endogenous cardiomyocyte generation could contribute to the benefits of exercise.

[1] Department of Stem Cell and Regenerative Biology and the Harvard Stem Cell Institute, Harvard University, Cambridge, MA 02138, USA. [2] Massachusetts General Hospital, Cardiology Division and Corrigan Minehan Heart Center, Boston, MA 02114, USA. [3] Harvard Medical School, Boston, MA 02115, USA. [4] Institut Curie, PSL Research University, INSERM, U1196, 91405 Orsay France. [5] Université Paris-Sud, Université Paris-Saclay, CNRS, UMR 9187, 91405 Orsay France. [6] Center for NanoImaging, Brigham and Women's Hospital, Cambridge, MA 02138, USA. [7] Department of Medicine, Division of Genetics, Brigham and Women's Hospital, Boston, MA 02115, USA. [8] Department of Biomedical Engineering, Case Western Reserve University, Cleveland, OH 44106, USA. [9] Department of Medicine, Division of Cardiovascular Medicine, Brigham and Women's Hospital, Boston, MA 02115, USA. These authors contributed equally: Ana Vujic, Carolin Lerchenmüller. These authors jointly supervised this work: Richard T. Lee, Anthony Rosenzweig. Correspondence and requests for materials should be addressed to R.T.L. (email: Richard_Lee@harvard.edu) or to A.R. (email: arosenzweig@partners.org)

Traditionally the adult mammalian heart was viewed as having negligible capacity for cardiomyogenesis. While we now recognize the adult heart has some regenerative capacity, its extent has been a topic of considerable controversy. Quantitation of cardiomyogenesis is challenging because cardiomyocytes are frequently polyploid and multinucleated[1] and many experimental approaches cannot distinguish generation of a second nucleus or ploidization from true mitotic events. Classical methods for studying cardiomyocyte proliferation use antibodies raised against mitotic markers and microscopy to visualize cell division. These methods are broadly applicable, but are susceptible to technical variability and present challenges for quantifying rare events such as a transient expression of cell cycle markers. A major benefit of multi-isotope imaging mass spectrometry (MIMS) is that it enables high-resolution quantification of prospectively-administered, non-radioactive stable isotope tracers, which do not interfere with biochemical reactions[2,3]. Importantly, we previously demonstrated that measurement of stable isotope tracers of DNA synthesis with MIMS enables quantitative discrimination between DNA repair and mitosis, as DNA repair incorporates orders of magnitude less thymidine than cell mitosis[3]. Multiple studies have used quantitative analyses of DNA isotope labeling to demonstrate a similar baseline rate of cardiomyogenesis in adult humans[4,5] and mice[3]. However, whether this basal level of cardiomyogenesis can be enhanced by physiological stimuli remains unclear.

This issue has considerable clinical importance since many cardiac diseases are associated with loss of cardiomyocytes in the adult heart[6,7]. In animal models, a remarkably modest attrition of cardiomyocytes (as few as 23 per 10,000 cells) is sufficient to induce fatal heart failure[8]. This is quantitatively similar to the number of apoptotic cardiomyocytes seen in human heart failure[9,10]. Thus, a modest imbalance between the loss and birth of cardiomyocytes in the adult heart could have profound implications for cardiac health, and continuous stimulation of cardiomyogenesis might counteract age-related loss of cardiomyocytes.

Repetitive exercise protects the heart from disease, with an impact similar to many drug interventions[11–14], although the basis for these benefits is incompletely understood. Regular exercise enhances cardiac function and increases myocardial mass, an effect that has historically been ascribed exclusively to increased cardiomyocyte size[15,16]. Expression profiling of exercised hearts identified an increase in transcription factors associated with cell cycle progression[17]. Cardiomyocyte markers of proliferation were also increased[17,18]. However, these studies did not account for ploidy and multinucleation, nor did they distinguish DNA repair from cell cycle activity; consequently this work was viewed as hypothesis-generating and inconclusive[19].

To definitively address whether exercise increases the birth of cardiomyocytes, here we study the effects of monitored voluntary exercise in mice with or without ischemic injury by analyzing genomic $^{15}$N-thymidine incorporation with MIMS, combined with cardiomyocyte nuclei tracing to assess nucleation and in situ hybridization to quantify ploidy. We find that voluntary running exercise significantly increases the number of $^{15}$N-thymidine labeled, mononuclear, diploid, cardiomyocytes in normal adult and injured mouse hearts, indicating exercise stimulates cardiomyogenesis. Mechanistically, we show that the exercise-induced increase in cardiomyocyte $^{15}$N-thymidine labeling can be abolished by inhibition of miR-222, a microRNA increased by exercise in both animal models and humans. We conclude that exercise stimulates cardiomyogenesis in the injured and uninjured adult mouse heart and that miR-222 is necessary for the cardiomyogenic response.

## Results

**Exercise enhances $^{15}$N-thymidine-positive cells in the heart.** We subjected young adult C57Bl/6 mice (2 months old) to 8 weeks of monitored voluntary wheel running (averaging $5.57 \pm 0.63$ km/day, mean $\pm$ s.e.m.) with continuous infusion of $^{15}$N-thymidine. Consistent with previous studies[17,20], exercised mice demonstrated physiologic cardiac remodeling characterized by increased cardiac mass (Supplementary Fig. 1a), augmented echocardiographic (Supplementary Fig. 1b), and cellular measures of hypertrophy (Supplementary Fig. 1c, d). Messenger RNA analysis also confirmed an increased alpha/beta myosin heavy chain (*MYH6/MYH7*) ratio consistent with physiological hypertrophy (Supplementary Fig. 1e). No signs of systolic dysfunction or increased apoptosis were detected in the hearts of exercised mice (Supplementary Fig. 2a, b). As previously described[2,3], we leveraged the sub-organelle resolution of MIMS, that allowed detection of characteristic ultrastructures, such as sarcomeres, to quantify incorporation of $^{15}$N-thymidine within cardiomyocytes (Fig. 1a). We analyzed 1209 cardiomyocytes from four sedentary mice compared with 1130 cardiomyocytes from four exercised mice. Cardiomyocyte nuclei that underwent DNA synthesis during the labeling period were identified by $^{15}$N labeling. These cells displayed a stereotypical pattern of peripheral heterochromatin and a distinct nucleolus (Fig. 1a). We observed a low frequency of cardiomyocyte DNA synthesis in sedentary control mice ($1.24\% = 0.02\%$ per day), consistent with our previous results[3]. However, exercised mice had a significantly higher fraction of $^{15}$N-thymidine labeled cardiomyocytes ($3.62\% = 0.06\%$ per day, $p = 0.0003$, Fisher's exact test, odds ratio (OR) = 2.997, confidence interval (CI) 1.65–5.45) (Fig. 1b, c). In conjunction with the increase in cardiomyocyte DNA synthesis, we also detected an increase in $^{15}$N-thymidine labeled non-cardiomyocytes (non-CMs) in exercised hearts compared to sedentary controls (Supplementary Fig. 2c). To determine if this increase in $^{15}$N-thymidine labeled non-CMs was accompanied by an exercise-induced neo-angiogenic response, we analyzed the capillary density by immunohistochemistry and found an increased number of capillaries per cardiomyocyte following exercise training ($4.14 \pm 0.07$ vs. $3.75 \pm 0.08$ capillaries/cardiomyocyte in exercised vs. sedentary controls, mean $\pm$ s.e.m., $p < 0.05$, two-sided $t$ test) (Supplementary Fig. 2d). These findings are consistent with exercise-induced angiogenesis, which has been previously described[21,22].

**Exercise induces cardiomyogenesis.** To account for the polyploidy and multinucleation that can be seen in cardiomyocytes[1,3], particularly during hypertrophic growth[23], we performed cardiomyocyte tracing on serial periodic acid Schiff (PAS)-stained sections in both directions to the $^{15}$N-thymidine-labeled cardiomyocyte nuclei to define the number of nuclei contained in each cell (Fig. 2a). We then utilized in situ hybridization targeting the Y-chromosome in adjacent sections to assess ploidy of each $^{15}$N-thymidine labeled cardiomyocyte nucleus (Fig. 2a). Although we identified polyploid ($\geq 4n$) and binucleated $^{15}$N-thymidine cardiomyocytes (Fig. 2b), we also observed a higher frequency of diploid/mononucleated $^{15}$N-thymidine-labeled cells in exercised hearts relative to those from sedentary mice (Fig. 2c), consistent with an exercise-mediated increase in cardiomyogenesis ($1.15\%$ vs. $0.25\%$, $p = 0.01$, Fisher's exact test, OR = 4.695, CI 1.44–15.53) (Fig. 2c). In line with previous findings[24], mononucleated $^{15}$N-thymidine-labeled cardiomyocytes were significantly smaller than their binucleated counterparts in

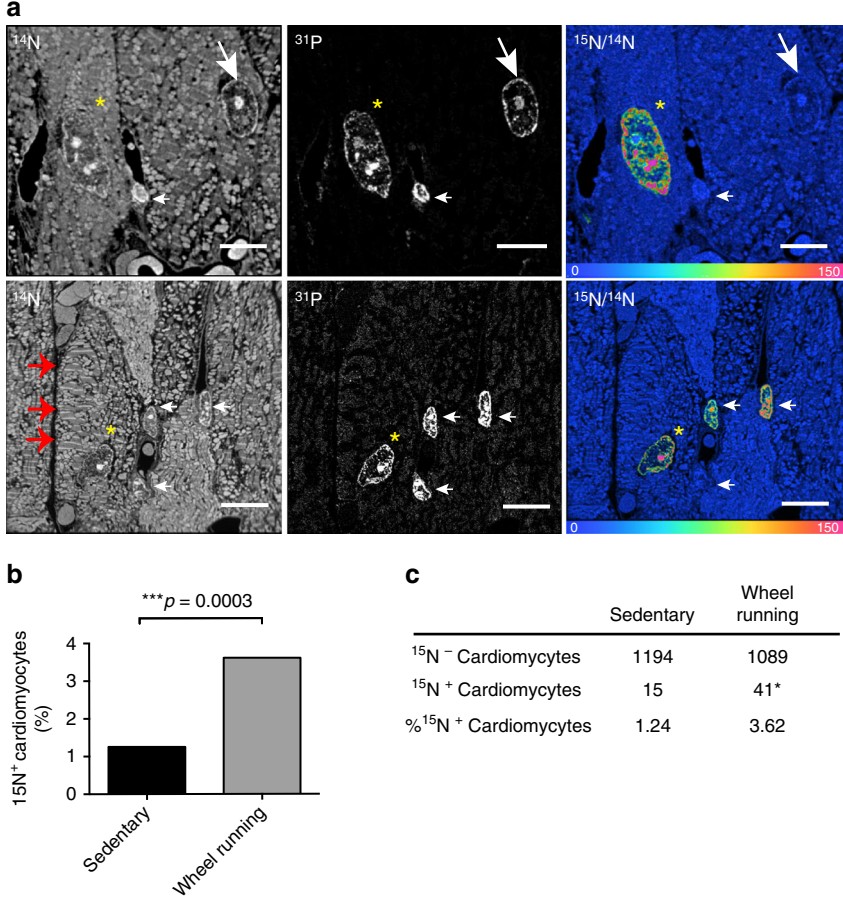

**Fig. 1** Cardiomyocyte cell cycle activity is increased in the exercised heart $^{15}$N-thymidine was administered continuously for 8 weeks to young adult mice (2 months old) undergoing voluntary wheel running vs. sedentary activity. **a** Mass $^{14}$N image (top left, bottom left) shows histological details such as sarcomeres (large red arrows), mass $^{31}$P image (center, top and bottom) shows nucleus and chromatin condensation, while the hue-saturation-intensity image (mosaic, top, and bottom right) demonstrates nuclear $^{15}$N labeling of a cardiomyocyte (yellow asterisk), non-labeled cardiomyocytes (large white arrows) can also be found in that section, as well as, two labeled non-cardiomyocytes (bottom right, small white arrows) and one non-labeled non-cardiomyocyte (top right, small white arrow). The scale ranges from blue, where the ratio is equivalent to natural ratio (0.37%, expressed as 0% above natural ratio (enrichment over natural ratio)), to red, where the ratio is 150% above natural ratio. $^{15}$N-thymidine has labeled the nucleus while the cytoplasm is at the natural abundance level. Scale bar = 10 μm. **b** Comparison of the percentage of $^{15}$N-labeled cardiomyocyte nuclei in exercised to sedentary young adult hearts. Exercise increases cardiomyocyte cell cycle activity (sedentary:exercise = 1.24:3.62%; >1000 cardiomyocytes from four mice per group were counted, ***$p = 0.0003$, Fisher's exact test). **c** Contingency table showing the absolute numbers and percentage calculations of $^{15}$N-positive and $^{15}$N-negative cardiomyocytes

the same hearts (Fig. 2d). Although rare, we also found $^{15}$N-thymidine labeled cardiomyocytes with a disturbed sarcomere structure and two adjacent nuclei that appeared to have completed cytokinesis at the time of tissue harvest (Supplementary Fig. 2e). Taken together, these results demonstrate that regular exercise over 8 weeks increased cardiomyogenesis. The numbers translate into a 4.6-fold increase in cardiomyogenesis in exercised animals (0.25% per 8 weeks = 0.0045% per day in sedentary vs. 1.15% per 8 weeks = 0.021% per day in exercised animals). These data suggest that the exercised heart is capable of generating new cardiomyocytes at a projected annual rate of 7.5% vs. 1.63% in sedentary conditions.

**Exercise after MI increases the area of proliferation**. Exercise mitigates adverse remodeling when initiated after myocardial infarction (MI) in animal models[25–27] and humans[28], attenuating fibrosis, dilatation, and cardiac dysfunction, although the mechanisms are incompletely understood. We sought to

determine whether exercise-induced cardiomyogenesis could contribute to the benefits of exercise after ischemic injury. Young adult mice underwent experimental MI by ligation of the left anterior descending artery followed by voluntary running or sedentary activity, started 24 h after ischemic injury with continuous labeling with $^{15}$N-thymidine for 8 weeks during the exercise regimen. We have previously shown that cardiomyocyte cell cycle activity increases in the peri-infarct region[3]. In this study, we confirmed this increase in the peri-infarct region in $^{15}$N-thymidine labeled cardiomyocytes in both sedentary (22.8%) and exercised (20.4%) mice, consistent with our previous data, but with no significant difference between the two groups (Fig. 3a–c). Distinct from the peri-infarct region, exercise led to a substantially higher proportion of $^{15}$N-thymidine labeled cardiomyocytes over 8 weeks in the extended border zone region (>400 μm from infarct) (19.1% $^{15}$N-thymidine labeled cardiomyocytes in exercised vs. 5.3% in sedentary mice, $p < 0.0001$, Fisher's exact test) (Fig. 3d). When examining the nucleation and ploidy status of $^{15}$N-thymidine labeled cardiomyocytes in this extended border region, we found a higher frequency of diploid/

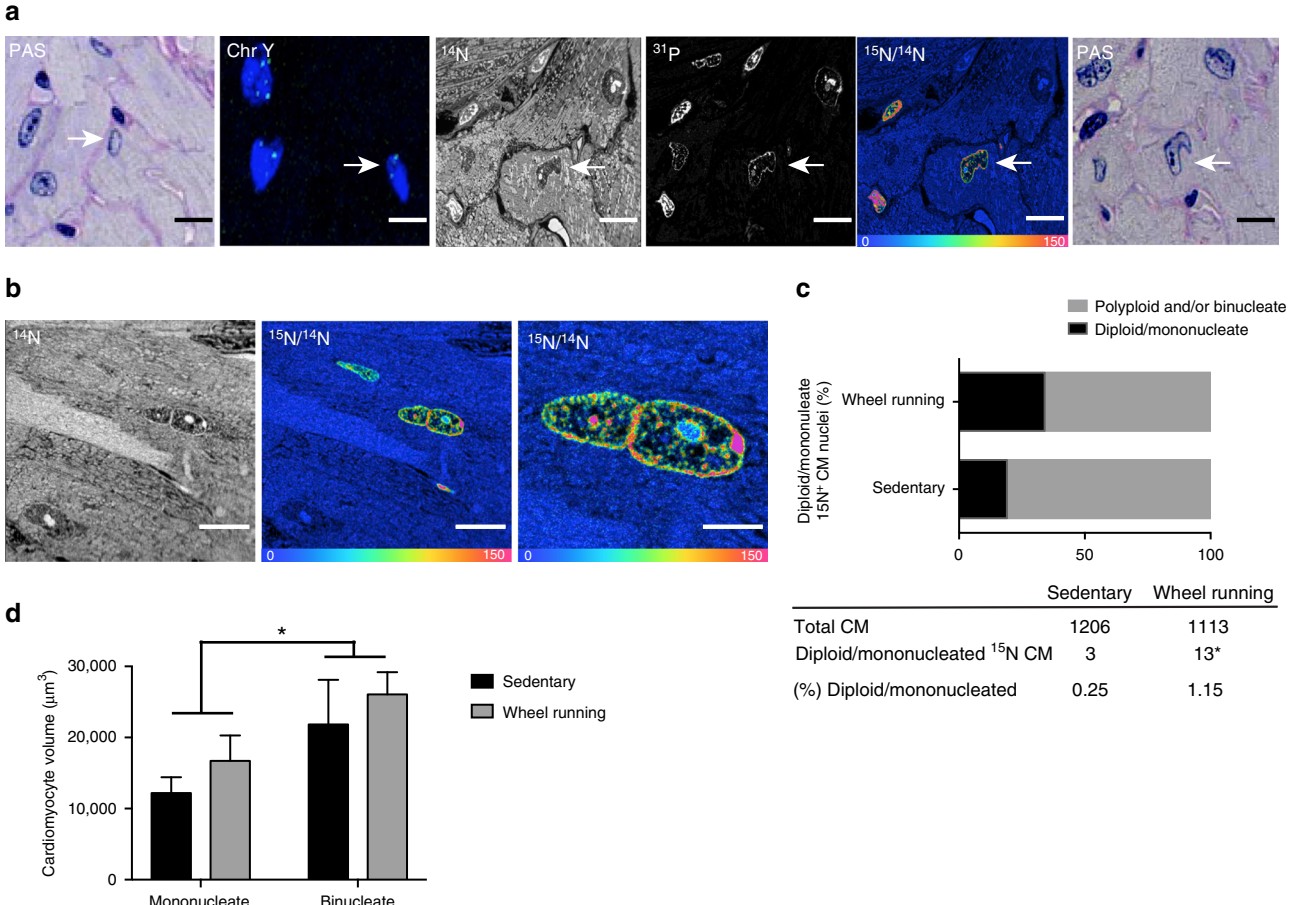

**Fig. 2** Number of mononucleate diploid $^{15}$N-labeled cardiomyocytes increases with exercise. **a** Serial sections (0.5–1 μm thickness) were processed to determine the ploidy status and number of nuclei of each $^{15}$N-labeled cardiomyocyte (large white arrow). Periodic acid Schiff staining (PAS) was performed on serial adjacent sections in both directions from the MIMS chip to define the number of nuclei contained in the cell and fluorescent in situ hybridization (Y-chromosome) was performed to identify ploidy status. A representative image series is shown from a mononucleate (PAS staining far left/right, scale bar = 10 μm) diploid (2 N, second from left, scale bar = 5 μm) $^{15}$N-labeled (MIMS second from right, scale bar = 10 μm) cardiomyocyte. $^{14}$N and $^{31}$P images are shown for subcellular resolution (center, scale bar = 10 μm). **b** Representative image of a $^{15}$N-labeled cardiomyocyte nuclei undergoing binucleation. Scale bar = 10 μm for mass image $^{14}$N and $^{15}$N/$^{14}$N (left, center). Scale bar = 4 μm for mass image $^{15}$N/$^{14}$N (right). **c** Bar graph showing the frequency of mononucleate/diploid vs. polyploid and/or multinucleate $^{15}$N-thymidine-labeled cardiomyocytes from each group (graph) and contingency table showing absolute numbers and percentage calculations of $^{15}$N-positive and all identified cardiomyocytes (sedentary: exercised = 0.25%:1.15%, $n = 4$ mice per group, *$p = 0.01$, Fisher's exact test, OR = 4.695, CI 1.44–15.53). **d** Mononucleate $^{15}$N-thymidine labeled cardiomyocytes were significantly smaller than their binucleate counterparts in the same hearts ($n = 3$ mice per group, one-way ANOVA with Tukey's correction for multiple comparisons (significance level $p < 0.05$). Error bars represent ± s.e.m

mononucleated $^{15}$N-thymidine-labeled cells in hearts from exercised compared with those from sedentary mice (Fig. 3e), consistent with an exercise-mediated increase in cardiomyogenesis (0.4% vs. 2.7% of total cardiomyocytes, $p = 0.004$, Fisher's exact test, OR = 6.931, CI 1.87–30.83) (Fig. 3f). The higher frequency of proliferating cells in the remote area, has also been observed after MI in mice exposed to hypoxia[29]. Transgenic activation of cardiomyocyte cell cycle activity in animals improves cardiac function after infarction[30]. Thus, the exercise-induced cardiomyocyte cell cycle activity seen after MI would be expected to favorably influence cardiac remodeling and likely contributes to the well-documented benefits of exercise in this setting[25–28].

**Inhibition of miR-222 blocks exercise-induced cardiac growth.** To explore the mechanism of the cardiomyogenic exercise response, we studied the role of miR-222, a microRNA (miRNA) that increases in response to exercise in both animal models and

humans, and plays an important role in the cardiovascular effects of exercise[18,31]. Although our prior work had suggested a role for miR-222 in cardiomyocyte proliferation[18], these studies were based on work in neonatal rather than adult cardiomyocytes, and proliferation markers that could not unambiguously identify mitotic events in vivo[18]. First, we confirmed that voluntary wheel running induced miR-222 expression in the adult mouse heart (Fig. 4a). Next, we inhibited miR-222 upregulation by weekly injection of a sequence-specific locked nucleic acid inhibitor (LNA)-anti-miR-222, together with $^{15}$N-thymidine administration during the 8-week voluntary exercise protocol. Exercised mice injected with scrambled LNA-anti-miR (LNA-Ctr) and sedentary mice treated with LNA-anti-miR-222 served as controls. After 8 weeks of voluntary wheel running, LNA-Ctr treated mice developed physiologic cardiac hypertrophy (Fig. 4b). The cardiac hypertrophic response was attenuated in exercised mice treated with LNA-anti-miR-222, confirming that miR-222 is necessary for exercise-induced cardiac growth even over longer duration (8 weeks) than previously examined[18]. To determine

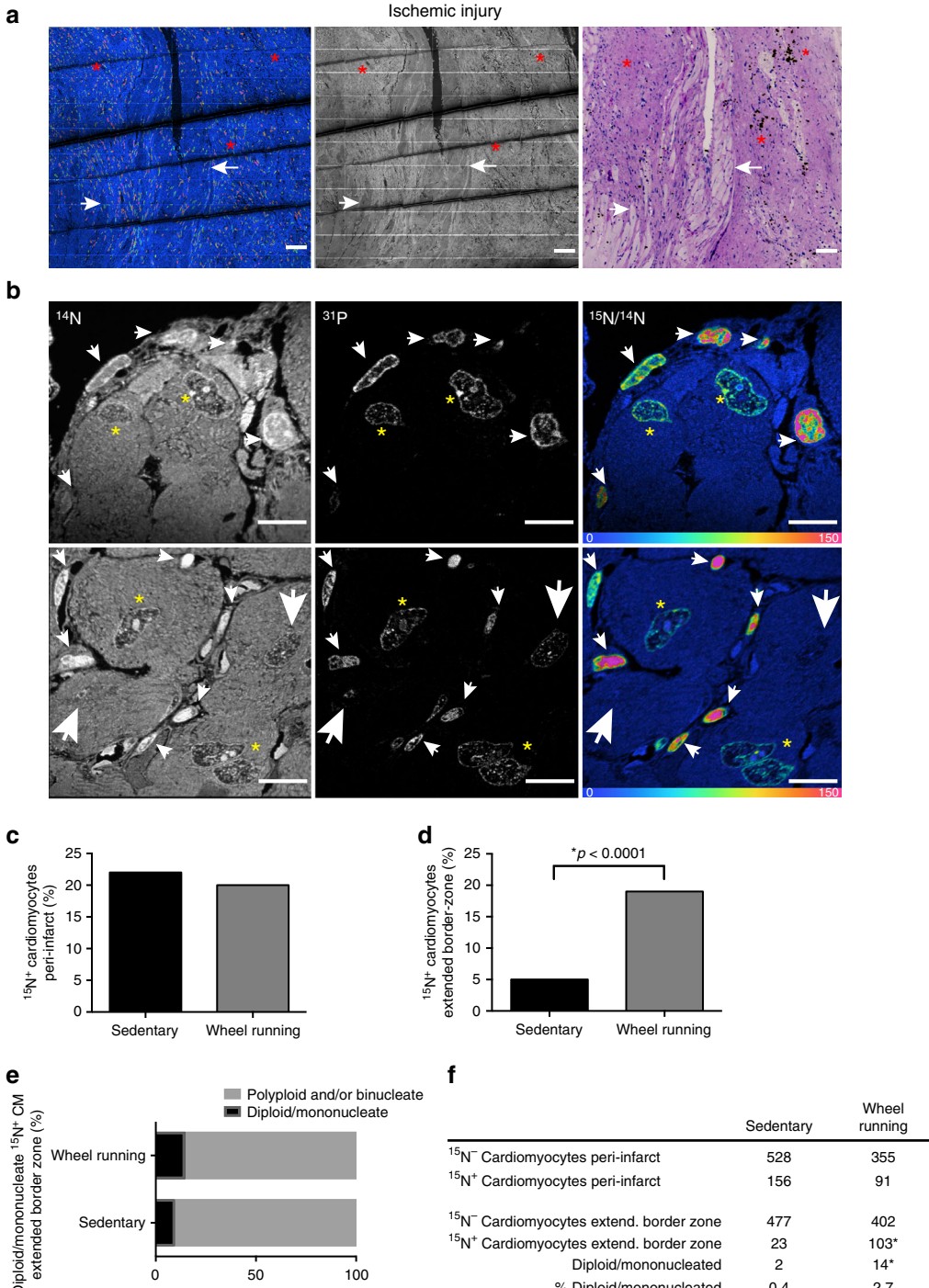

**Fig. 3** Exercise induces cardiomyogenesis in an extended MI border zone. Mice were subjected to experimental myocardial infarction (MI) by ligation of the left anterior descending artery and exposed to exercise or sedentary activity for 8 weeks, starting 24 h post surgery. Subcutaneous osmotic pumps were implanted to continuously label mice with $^{15}$N-thymidine for eight weeks following surgery. **a** Myocardial ischemic injury results in extensive DNA synthesis in the peri-infarct area. $^{15}$N:$^{14}$N hue-saturation-intensity image (HSI mosaic) (left) demonstrates $^{15}$N$^{+}$ cells while $^{14}$N mass image (center) and PAS staining (right) demonstrate presence of cardiomyocytes (white arrow) and a visible scar/fibrosis (dark purple PAS staining, red asterisks). The HSI mosaic scale ranges from blue, where the ratio is equivalent to natural ratio (0% above natural ratio (enrichment over natural ratio)), to red, where the ratio is 150% above natural ratio. Scale bar = 60 μm. **b** Representative magnifications from the peri-infarct and extended border zone areas. Mass $^{14}$N image (top left, bottom left), mass $^{31}$P image (center, top, and bottom), and the HSI mosaic (top right, bottom right) demonstrate nuclear $^{15}$N labeling of cardiomyocytes undergoing DNA synthesis (yellow asterisk), non-labeled-cardiomyocytes (large white arrows), and $^{15}$N-labeled non-cardiomyocytes (small white arrows). Scale bar = 56 μm. $^{15}$N-thymidine has exclusively labeled the nucleus while the cytoplasm is at the natural abundance level. **c** Percentage of $^{15}$N$^{+}$ cardiomyocyte nuclei after MI with or without exercise in the peri-infarct region (sedentary:exercise = 22.76%:20.43%; >400 cells from three mice per group were counted, $p$ = ns, Fisher's exact test) and **d** the extended border zone of the infarct (sedentary: exercise = 5.29%:19.09%; >500 cells from three mice per group were counted, $p < 0.0001$, Fisher's exact test). **e** Bar graph showing the frequency of mononucleate/diploid vs. polyploid and/or multinucleate $^{15}$N-thymidine-labeled cardiomyocytes from each group. **f** Contingency table showing the absolute numbers of $^{15}$N-labeled mononucleate/diploid cells of total counted cardiomyocytes from each group (sedentary:exercise = 0.4%:2.7%, $p = 0.004$, Fisher's exact test, OR = 6.931, CI 1.87–30.83)

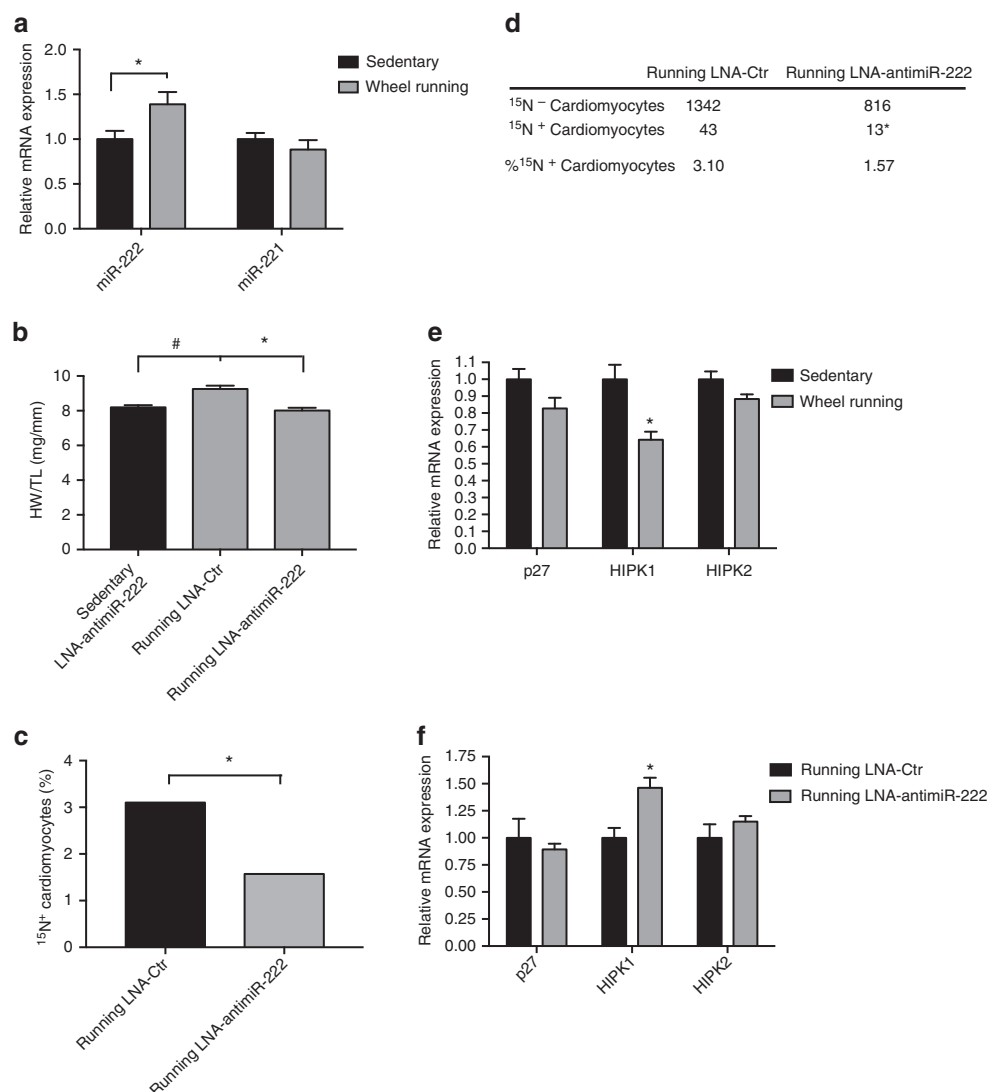

**Fig. 4** Inhibition of miR-222 prevents exercise-induced cardiomyogenesis. **a** miR-222 is upregulated after eight weeks of voluntary wheel running in young adult mice ($n = 6$ mice per group, $*p = 0.01$, Student's $t$ test). **b** Mice underwent simultaneous [15]N-thymidine infusion and LNA-anti-miR-222 or control LNA-anti-miR (LNA-Ctr) treatment for 8 weeks of sedentary activity or voluntary wheel running. miR-222 inhibition blocks physiologic cardiac hypertrophy measured by heart weight/tibia length (HW/TL) ($n = 5$ mice per group, $*p < 0.05$ running LNA-anti-miR-222 vs. running LNA-Ctr, $\#p < 0.05$ running LNA-Ctr vs. sedentary LNA-anti-miR-222, one-way ANOVA with Tukey's post-test for multiple comparisons). **c** Similar to exercise only, exercised mice injected with LNA-Ctr for 8 weeks show an increase in [15]N-thymidine-positive cardiomyocytes. However, exercised mice treated with LNA-anti-miR-222 demonstrate a reduced number of [15]N-thymidine-positive cardiomyocytes closer to sedentary baseline levels (800–1350 cells from four mice per group were counted $*p = 0.0255$, Fisher's exact test). **d** Contingency table showing the absolute numbers and percentage calculations of [15]N-positive and [15]N-negative cardiomyocytes. **e** Exercised mouse hearts show downregulation of miR-222 target HIPK1. Bar graph depicting quantitative results from gene expression analysis from heart lysates after 8 weeks of voluntary wheel running demonstrates significant downregulation specifically of HIPK1 ($n = 3$ mice per group, $*p < 0.05$, Student's $t$ test). **f** miR-222 inhibition during 8 weeks of voluntary wheel running leads to HIPK1 overexpression ($n = 5$ mice per group, $*p < 0.05$, Student's $t$ test). Error bars represent ± s.e.m

whether miR-222 also mediates exercise-induced cardiomyogenesis, we compared the frequency of [15]N-thymidine labeled cardiomyocytes in exercised hearts after treatment with either LNA-anti-miR-222 or LNA-Ctr (Fig. 4c, d). Interestingly, miR-222 inhibition fully prevented the increase in [15]N-thymidine incorporation in response to exercise (Fig. 4c, d). miR-222 inhibition also led to a more modest induction in exercise-induced [15]N-thymidine-positive non-CMs (Supplementary Fig. 3).

To further explore the mechanisms by which miR-222 regulates exercise-induced cardiomyogenesis, we performed quantitative gene expression analysis from sedentary and exercised mouse hearts for relevant genes previously validated as direct targets of miR-222[18]. Our data demonstrate that after 8 weeks of endurance exercise, HIPK1 gene expression remained significantly inhibited, without a significant change in p27 or HIPK2 expression (Fig. 4e). Moreover, HIPK1 expression increased significantly with miR-222 inhibition (Fig. 4f) (which abolished exercise-induced cardiomyogenesis) while p27 and HIPK2 expression did not. Taken together with our previously published data demonstrating that HIPK1 is a direct target of miR-222 with anti-proliferative effects in cardiomyocytes[18], these data strongly suggest that HIPK1 contributes to miR-222's modulation of exercise-induced cardiomyogenesis.

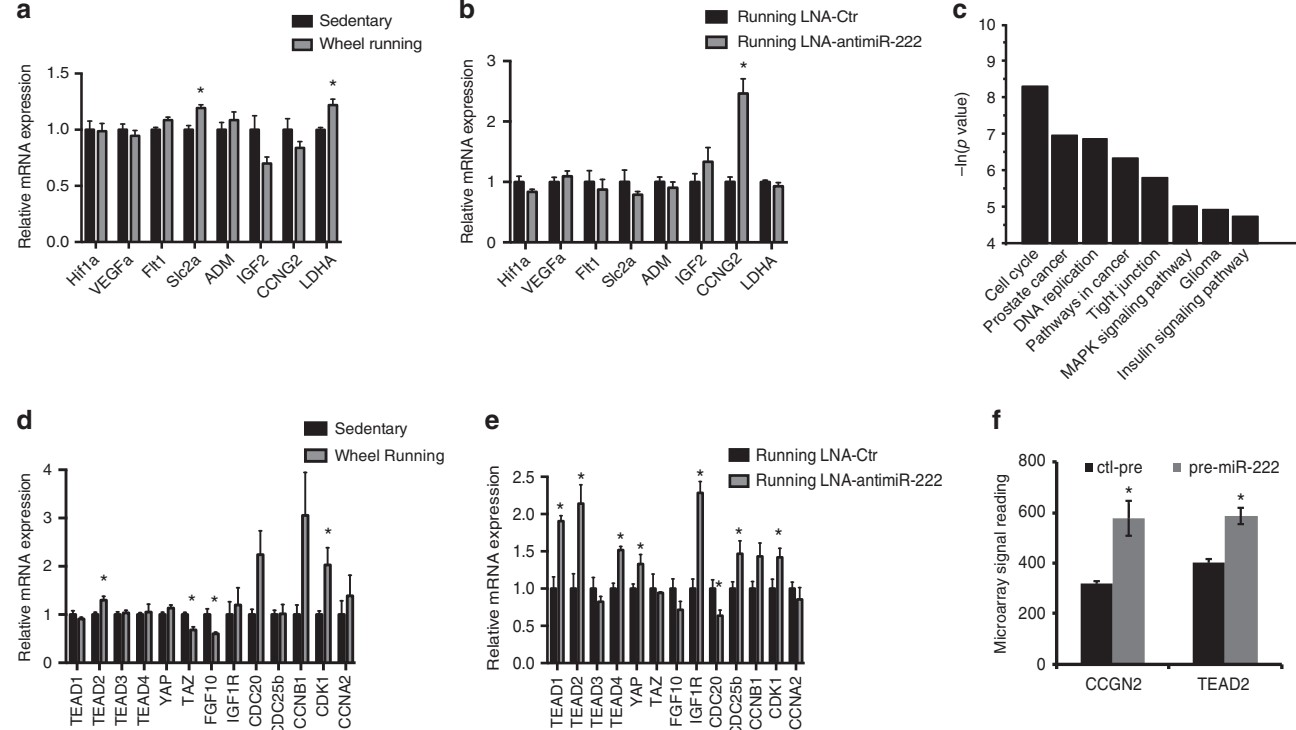

**Fig. 5** Hypoxia and hippo pathways are not directly targeted by miR-222. **a**, **b** qPCR was used to analyze mRNA levels of genes relevant for hypoxia-induced effects in exercised vs. sedentary hearts (**a**) and hearts from mice treated with LNA-anti-miR222 or LNA-scr-miR undergoing voluntary wheel running (**b**) ($n = 3$–5 mice per group, *$p < 0.05$, Student's $t$ test). **c** $p$ value from significantly differential pathways analyzed by ingenuity pathway analysis (IPA, Qiagen) of a microarray conducted from neonatal rat ventricular myocytes (NRVM) treated with control precursor (ctl-pre) and miR-222 precursor (pre-miR-222), respectively ($n = 4$ individual samples per group). **d**, **e** qPCR was used to analyze mRNA levels of genes relevant for hippo pathway-induced effects in exercised vs. sedentary hearts (**d**) and hearts from mice treated with LNA-anti-miR222 or LNA-scr-miR undergoing voluntary wheel running (**e**) ($n = 3$–5 mice per group, *$p < 0.05$, Student's $t$ test). **f** Gene expression signal specifically for CCGN2 and TEAD2 detected in a microarray conducted from neonatal rat ventricular myocytes (NRVM) treated with control precursor (ctl-pre) and miR-222 precursor (pre-miR-222), respectively ($n = 4$ individual samples per group, *$p < 0.05$, Student's $t$ test). Error bars represent ± s.e.m

We further investigated other pathways recently linked to cardiomyocyte proliferation. For example, we examined a panel of hypoxia-induced genes[32,33], and found an increase in *Slc2a* (GLUT1) and *LDHA* expression after 8 weeks of exercise, pointing to a significant metabolic adjustment potentially due to increased Hif1a activity (Fig. 5a). However, these changes were not significantly affected by miR-222 inhibition suggesting they do not contribute to miR-222's modulation of cardiomyogenesis (Fig. 5b). In contrast, we did find that miR-222 inhibition increased expression of *CCNG2* (cyclin G2), a relevant cell cycle regulator, but could not find *CCNG2* as a predicted miR-222 target (through Pictar and Targetscan). We went on to specifically check *CCNG2* expression in miR-222 overexpressing neonatal rat ventricular cardiomyocytes (NRVMs), and found an increase in *CCNG2* expression, strongly suggesting that this is not a direct miR-222 target (Fig. 5c).

Cell cycle regulators that are reportedly downstream effectors of the Hippo pathway, another pathway implicated in cardiomyocyte proliferation, were increased in exercised hearts. While *CDK1*, *CCNB1*, and *TEAD2*, for example, were all upregulated with exercise (Fig. 5d), miR-222 inhibition caused further increase of those Hippo/cell cycle regulators (Fig. 5e). We subsequently ran a pathway analysis (Ingenuity Pathway Analysis, IPA, Qiagen) from a microarray comparing NRVM with precursor-miRNA-mediated miR-222 overexpression to controls (pre-miR-222 and Ctr-pre) and found cell cycle regulators in general, and with subsequent analysis (Fig. 5f), *TEAD2* specifically, to be upregulated with miR-222

expression, suggesting that they are not direct targets of miR-222's effects in this context. Further exploration of the role of the Hippo pathway in endurance exercise although apparently independent of miR-222 may be of interest for future studies.

## Discussion

Our data collectively demonstrate that regular running exercise stimulates an increase in new cardiomyocyte formation, and support the idea that exercise can activate the adult mammalian heart's endogenous capacity to regenerate. Moreover, these studies indicate that miR-222, and potentially other miR-222-independent cell cycle regulators, are necessary for exercise-induced cardiomyogenesis and suggest that enhanced cardiac regeneration may contribute to the benefits of exercise after myocardial injury. Our observations raise the intriguing possibility that this is a more general effect of exercise, and that identifying the signals responsible could have therapeutic implications. Repeated exercise could favorably shift the balance between cardiomyocyte birth and loss. Although the extent of cardiomyogenesis remains limited, the relative increase is significant (~4.6-fold) and the number of new cardiomyocytes formed is much greater than the modest number of apoptotic cardiomyocytes sufficient to cause lethal heart failure[8] and thus seems likely to influence these outcomes. However, one limitation of this study is that the exercise regimen the mice underwent is difficult to compare to human exercise scenarios. Understanding

the long-term impact of exercise on cardiomyocytes and responsible mechanisms may provide new insights into cardiac regeneration and new opportunities for intervention.

## Methods

**Mice**. All mice were maintained and studied using protocols in accordance with the Guide for the Use and Care of Laboratory Animals and approved by Harvard University (protocol number 16-05-273) and Massachusetts General Hospital (protocol number 2015 N000029) Animal Care and Use Committees. Two-month-old male C57Bl/6 mice were obtained from Charles River.

**Running exercise protocol and [15]N-thymidine labeling**. Starting at an age of 2 months, osmotic minipumps (Alzet) were implanted subcutaneously, delivering [15]N-thymidine (Cambridge Isotopes) at a rate of $20 \mu g \, h^{-1}$ and exchanged weekly for 8 weeks. Mice were individually housed in plexiglass cages ($36 L \times 20 W \times 15 H$ cm) that contained a stainless-steel running wheel (diameter 11.4 cm; Mini-Mitter, Starr Life Science, USA) equipped with a tachometer. Mice ran voluntarily. Mouse activity and cardiac hypertrophy response was controlled for by recording daily running and morphometric analysis. The sedentary control mice were kept in the same cage system lacking running wheels.

**Experimental myocardial infarction**. Mice were subjected to experimental MI. Surgeries were performed by a single operator with >20 years of experience in the performance of coronary ligation in rodents. In brief, the animals were anesthetized with isoflurane, intubated, and ventilated. After the thoracic cavity was opened, the left anterior descending coronary artery was permanently ligated approximately 2 mm below the left atrial appendage. The chest was closed and an osmotic mini-pump (Alzet) administrating [15]N-thymidine (Cambridge Isotope Laboratories) at a rate of $20 \mu g \, h^{-1}$, was implanted subcutaneously and the animal was placed in recovery. The pump was exchanged weekly for 8 weeks.

**miR-222 inhibition**. LNA-anti-miR injections were performed as previously described[18]. Two-month-old C57Bl/6 male mice were subcutaneously injected with $10 \, mg \, kg^{-1}$ of LNA-modified anti-miR-222 (LNA-anti-miR-222) or scrambled control (LNA-Ctr) reconstituted in saline for three consecutive days after osmotic pump implantation and then weekly after pump exchanges throughout the experiment. LNA-anti-miR oligonucleotides were purchased from Exicon.

**Echocardiographic studies**. Echocardiography was performed on conscious mice by using a GE Vivid E90 with a L8-18i-D probe. Parasternal long-axis views, short-axis views, and two-dimensional-guided M-mode images of short axis at the papillary muscle level were recorded. The average of at least three measurements was used for every data point from each mouse.

**Sample preparation and MIMS data acquisition and analysis**. Tissues were fixed with 4% paraformaldehyde (PFA), embedded in LR (London resin) white, sectioned (0.5–1 μm), and mounted on silicon chips. MIMS analyses were conducted with ion microscopy (NanoSIMS 50 and a large-radius NanoSIMS 50L, Cameca) as described previously[2,3]. Briefly, a focused $Cs^+$ ion beam (size <100 nm) is rastered over the sample surface. The impact of the $Cs^+$ ion beam on the sample surface generates the sputtering of multiple secondary ions from the probed nanovolume (corresponding to a pixel in the image). For each nanovolume, the secondary-ion intensities for $^{12}C^-$, $^{12}C^{14}N^-$, $^{12}C^{15}N^-$, and $^{31}P^-$ were recorded in parallel. The detection of nitrogen requires the use of poly-atomic ions $^{12}C^{14}N^-$ and $^{12}C^{15}N^-$ as a proxy for $^{14}N$ and $^{15}N$. A hue-saturation-intensity transformation of the $^{12}C^{15}N^-/^{12}C^{14}N^-$ ratio image provides a visual representation of regions where the ratio is above natural abundance (natural $^{15}N/^{14}N$ 0.37%), indicative of [15]N-thymidine labeling. Observers blinded to group assignment performed cardiomyocyte counts as well as the following analyses.

**Nucleation and cell volume analysis**. Serial sections (0.5–1 μm) were cut from LR white embedded heart to cover up to 100 μm of tissue below and above the section that was mounted on the chip. Sections were stained with a modified PAS staining protocol to identify individual cardiomyocytes. Slides were immersed for 2 × 30 min in xylene and rehydrated through graded alcohols, incubated in Periodic acid solution overnight and then in Schiff's reagent for two nights, washed, and dehydrated and cleared before mounting. Slides were imaged on a NanoZoomer Whole Slide Scanner (Hamamatsu Photonics). Magnified PAS image in Fig. 3a was taken on Airy Scan LSM800 with Axiocam color camera (Zeiss). To analyze nucleation, [15]N-thymidine-positive cardiomyocytes were tracked by locating the cardiomyocyte in every section of the serial above and below the level of the nucleus for as long as it was present. The total number of nuclei was counted for each cell. PAS-stained sections were also used to confirm cells labeled as cardiomyocytes and non-CMs in all cohorts and adjustments have been made for mis-labeled cells if necessary. Outlined [15]N-positive cells were also used to calculate total cellular volume.

**Fluorescent in situ hybridization**. Sections were incubated in 1 M sodium thiocyanate for 10 min at 80 °C, washed in PBS, and treated for 2 min in 0.2% Triton X (in PBS), digested with Proteinase K ($50 \mu g \, ml^{-1}$) for 30 min at 56 °C. Sections were rinsed in 1× PBS, post-fixed in 4% PFA/PBS for 5 min and treated for 10 min with 50% formamide/4× standard saline citrate buffer at 37 °C. Slides were air-dried and biotinylated-labeled chromosome Y probe (IDMF 1057, Empire Genomics) was suspended in hybridization mix, applied to sections, and sealed under glass with rubber cement. Samples were denatured at 69 °C for 3 min. After overnight incubation at 42 °C, slides were washed three times with 2× standard saline citrate buffer, two 10 min washes each with PBS containing 0.1% Tween (PBST), all at room temperature. Samples were blocked with PBST/10% goat serum and incubated for 1 h in streptavidin-conjugated Alexa Fluor 488 (S32354, Invitrogen) before being washed and mounted. Sections were imaged on a LSM 510 Inverted Confocal (Zeiss LSM 510) and analyzed with Zen lite software (Zeiss).

**Capillary density and cross-sectional area quantification**. Hearts from animals that underwent the above described exercise regimen but that were not used for LR white embedding and MIMS were snap frozen in OCT (optimal cutting temperature) as previously described[18]. Thick sections of 6 μm were cut from the frozen tissue and placed on a glass cover slip, air-dried for 5 min, followed by a brief 10 min fixation with fresh 4% PFA. Sections were immunofluorescently stained with both CD31 (BD Pharmingen) and wheat germ agglutinin (488-oregon green, Thermo Fisher Scientific) to analyze capillary density (capillaries/cardiomyocyte) and cross-sectional area using standard procedures. AlexaFluor (555-texas red) fluorescent dye (Thermo Fisher Scientific) was used as secondary antibody. Images were taken on a Leica SP8 confocal microscope.

**Cell death assay by TUNEL staining**. Slides were rehydrated through xylene and graded concentrations of ethanol and stained for cell death using the TMR (tetramethylrhodamine)-red TUNEL (terminal deoxynucleotidyl transferase dUTP nick end labelled) Kit (in situ cell death Detection Kit-TMR Red; Roche, Basel, Switzerland) as per manufacturers instruction. Briefly, slides were incubated for 30 min at 37 °C in $20 \mu g \, ml^{-1}$ proteinase K, washed twice in 1× PBS 5 min at room temperature and incubated at 37 °C in a red fluorescent (TMR-red) TUNEL cell death detection reagent. Cardiomyocytes were identified by simultaneous immunostaining cardiac troponin T (1:200 dilution, ab8295, Abcam, Cambridge, MA). After washing with 1× PBST (1× PBS added with 0.1% Tween), slides were counterstained with DAPI (4′, 6-diamidino-2′-phenylindol, dichloride; Thermo Scientific/Pierce, Rockford, IL, USA) to visualize the nuclei. Finally, slides were washed and mounted and all quantitative analyses were performed. All images were post-processed in ImageJ (NIH) software. Approximately 14,000 troponin-T-positive cardiomyocytes were counted for the TUNEL analysis. LR = London Resin OCT = optimal cutting temperature TMR = Tetramethylrhodamine TUNEL = Terminal deoxynucleotidyl transferase dUTP nick end labelled

**RNA isolation and quantitative real-time PCR**. RNA was isolated from both tissue and cell samples using Trizol (Qiagen), and phase separation followed by column purification (Zymo Research). RNA of 1 μg was added to reverse transcription reactions (Applied Bioscience) and quantitative PCR was carried out using SYBR-green (Bio-Rad).

**Cardiomyocyte isolation and culture and gene expression**. Primary NRVMs were prepared from 1- to 2-day-old rats by use of the neonatal cardiomyocyte isolation system (Worthington Biochemical Corp.). Isolated NRVMs were purified by Percoll gradient centrifugation and plated in 60 mm dishes at $1 \times 10^6$ cells per well and cultured in DMEM/5% FBS/10% horse serum for 24 h. Before treatment, NRVM were synchronized, and cultured in 0.2% FBS DMEM media. miRNA-222 as well as control precursors (pre-miR) were purchased from Invitrogen. Transfection of pre-miR was carried out using Lipofectamine RNAiMax (Invitrogen) according to the manufacturer's protocol. Forty-eight hours after transfection, total RNA was isolated from treated NRVM with Trizol (Invitrogen) according to standard protocols and submitted to the Dana-Farber Cancer Institute Molecular Diagnostics Laboratory for microarray assay using Affymetrix Rat Genome 230 v2.0. Microarray data analysis was performed with Affymetrix Microarray Data Analysis tools.

**Statistical analysis**. Statistical testing was performed using Prism 3.0/7.0 (Graphpad). For echocardiographic and morphometric analyses, as well as gene expression, cross-sectional area, capillary density, and cell death/volume, results are presented as mean ± s.e.m. from $n = 3–5$ animals per group and were compared by using two-sided $t$ tests (significance level $p < 0.05$). More than two groups were compared by using one-way analysis of variance with Tukey's correction for multiple comparisons. Data comparing event rates, i.e., [15]N-thymidine-positive and -negative cardiomyocytes and non-CMs, were compared using Fisher's exact test based on the total number of cells counted as unit of analysis (significance level $p < 0.05$) and are presented as percentage of total

cell count as well as in a table. Three to four mice per group were used to ensure sufficient cell counts.

**Data availability**. The datasets generated during and/or analyzed during the current study are available from the corresponding author upon reasonable request. Microarray data have been deposited in the Gene Expression Omnibus (GEO accession number: GSE59641).

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

## Acknowledgements

We thank J. Gannon and C.Y. Xiao for surgical assistance, Y. Iwamoto and D. Capen of the Center for Systems Biology at Massachusetts General Hospital for pathology and histology services, and C. Heß and K. Mühlburger for technical assistance. C.L. was funded by the German Research Foundation (LE 3257 1-1). R.T.L. was funded by the NIH (HL117986, HL119230, and HL122987) and the Leducq Foundation. A.R. was funded by the NIH (HL122987, HL135886, and TR000901) and the AHA (14CSA20500002 and 16SFRN31720000). Confocal microscopy was partly performed in the Microscopy Core of the Center for Systems Biology/Program in Membrane Biology, which is partially supported by an Inflammatory Bowel Disease Grant DK043351 and a Boston Area Diabetes and Endocrinology Research Center (BADERC) Award DK057521. The microscope used in this facility is associated with grant #1S10OD021577-01.

## Author contributions

A.V., C.L., R.T.L., and A.R. designed the experiments. A.V. and C.L. performed the experiments. T-D.W. and J.-L.G.-K. conducted the NanoSIMS analyses at Institut Curie. C.G. and M.L.S. conducted NanoSIMS analyses at the Brigham and Women's Hospital Center for NanoImaging. A.V. and C.L. analyzed data and discussed analyses and results with R.T.L. and A.R., C.P.R. and E.G. supported data analyses. S.E.S. and X.L. provided critical feedback and support. A.V. and C.L. made the figures. A.V., C.L., R.T.L., and A.R. wrote the manuscript. All authors approved the manuscript.

## Additional information

**Competing interests:** The authors declare no competing interests.

