## [Peer Review File · Nature Communications]

Reviewer #1:

Remarks to the Author:

NCOMMS-17-19078

Exercise induces new cardiomyocyte generation in the adult mammalian heart

This paper evaluates the effect of running exercise on DNA synthesis in mononuclear, diploid cells as a surrogate for cell cycle activation following myocardial infarction using state of the art multi-isotope imaging mass spectrometry (MIMS). A landmark paper by the authors previously used this technology to reveal that very few new myocytes are created in adult mice, but that those that do form arise from cell division of pre-existing cardiomyocytes (Senyo et al., Nature, 2013). In this study, the authors focus on the influence of running exercise initiated after myocardial infarction. They conclude that the benefits include increased new myocyte creation and that the effect is attenuated when endogenous miR-222 is blocked with a locked nucleic acid anti-miR. The finding that exercise boosts new myocyte creation would have profound implications. However, the conclusion of cell division seems hard to distinguish from DNA repair or other mechanisms causing incorporation of thymidine into DNA. This problem undermines the overall conclusion of how exercise might be beneficial and it needs to be resolved.

Major issues:

1. Fig. 2. The conclusion stated on bottom of page 4 and continuing on to top of page 5 that exercised hearts substantially increase cardiomyogenesis seems based on the data showing that there was a trend, lacking statistical significance, towards an increase in the frequency of ¹⁵N labelled, diploid mononucleated cardiomyocytes. These cells are presumed to be cells that have incorporated the thymidine that carried the ¹⁵N label and remained diploid and mononucleated, so hence had not undergone polyploidization or binucleation. I believe that the authors inferred that these cells, by default, must have undergone cell division and for this reason state that there has been an increase in cardiomyogenesis. This is a major claim of the paper. However, if I understand correctly, it is based on a non-significant trend coupled to the absence of evidence for polyploidization or endoreduplication. On balance, this evidence seems rather weak for such a profound conclusion. I think it needs to be either much better explained (perhaps I am missing something) or better substantiated. Furthermore, the numbers are expressed as percentages rather than the actual number tallied, and it would have been useful to see the actual numbers presented.

2. Fig. 3. This figure examines the localization of ¹⁵N incorporation relative to the infarct region. Interestingly, an increased proportion of ¹⁵N incorporation detected >400 microns from the infarct zone in exercised animals vs sedentary animals. These data do not seem to have been subjected to the analyses as in Fig. 2, in which cells were discriminated by diploid, mononuclear cells to identify those that are candidates for cells that have undergone cell division. Thus, it seems incorrect to term

these cells as having undergone “cardiomyocyte cell cycle activity” (as stated on the first paragraph on page 5). It seems more accurate to characterize these cells as having undergone thymidine incorporation, which could be due to DNA damage, polyploidization, endoreduplication, etc.

3. Fig. 4. The results with LNA to miR-222 are impressive. Interestingly, the authors suggest on top of page 6 that one of the benefits of exercise might be “enhanced repair”. It is puzzling why this is the conclusion here, but not for the experiments in Fig. 3. The difficulty in separating repair from cell division mechanisms is a problem in this paper that needs to be resolved.

Reviewer #2:

Remarks to the Author:

This is an interesting and potentially important manuscript from two leading Harvard groups. This work builds on previous studies by the Rosenzweig lab on the effect of exercise, and the Lee lab using the MIMS approach to determine the rate of cardiomyocyte turnover. The manuscript outlines the cardiomyocyte turnover dynamics following 8 weeks of wheel exercise. The authors demonstrate clearly that there is in fact an increase in cardiomyocyte turnover with exercise. Although this increase in turnover rate is modest, it is an important finding that is worth pursuing. There are, however, several major concerns:

1) The implication of exercise in cardiomyocyte proliferation is intriguing in light of recent reports by the Belmonte group in Zebrafish, and the Sadek group in mice outlining the effect of hypoxia on cardiomyocyte proliferation. As such, the authors need to determine whether exercise activates a hypoxic stress response pathway in cardiomyocytes. Similarly, other critical pathways that have been implicated in cardiomyocyte proliferation such as the Hippo pathway, or Neuregulin need to be explored. As it stands, the mechanistic aspect of the current manuscript is thin.

2) The implication of miR222 is of interest, however the studies as they stand are somewhat superficial. What are the miR222 targets that mediate this observed effect on cardiomyocyte proliferation? does it regulate any of the aforementioned pathways? Importantly, if miR222 is required for the observed effect, does miR222 overexpression result in cardiomyocyte proliferation? This is a critical point in light of a recent report (Su et al, Cell Physiol Biochem 2016;39:1503-1511) which indicates that miR222 overexpression results in heart failure.

3) The findings of the spatial distribution of new cardiomyocytes is of interest. If I understand this correctly, the authors did not observe an increase in cardiomyocyte proliferation in the border zone post MI. If this is the case, this would be contrary to what is known about post-MI induction of cell cycle entry of cardiomyocytes in the border zone. In fact, I believe that the Lee group showed in the

original Nature report using the MIMS approach, that turnover in the borderzone is enhanced. This is an important point that needs clarification. If there is no increase in the borderzone, this needs to be explained experimentally.

4) As it stands, the conclusions are a bit inflated. For example, given that humans usually do not maintain such a rigorous exercise program for years, it is unclear how impactful an absolute increase in cardiocyte proliferation by 2% would achieve. For example, statements in the abstract such as "powerful physiological regulator" and "stimulation of cardiomyocyte proliferation likely contributes to the benefit of exercise" are not supported by the current findings.

Reviewer #1

Reviewer #1's Comment

“This paper evaluates the effect of running exercise on DNA synthesis in mononuclear, diploid cells as a surrogate for cell cycle activation following myocardial infarction using state of the art multi-isotope imaging mass spectrometry (MIMS). A landmark paper by the authors previously used this technology to reveal that very few new myocytes are created in adult mice, but that those that do form arise from cell division of pre-existing cardiomyocytes (Senyo et al., Nature, 2013). In this study, the authors focus on the influence of running exercise initiated after myocardial infarction. They conclude that the benefits include increased new myocyte creation and that the effect is attenuated when endogenous miR-222 is blocked with a locked nucleic acid anti-miR. The finding that exercise boosts new myocyte creation would have profound implications. However, the conclusion of cell division seems hard to distinguish from DNA repair or other mechanisms causing incorporation of thymidine into DNA. This problem undermines the overall conclusion of how exercise might be beneficial and it needs to be resolved.”

We thank the reviewer for his/her comments on the potential importance of our study. We agree that distinguishing DNA repair from cell cycle events is critically important and an important rationale for using MIMS analysis, since it is a quantitative assay that allows us to distinguish these processes. As noted above (please see answer #1 to Editor's comments), the amount of thymidine incorporation is dramatically lower in repair than mitosis, and previously published work from our group demonstrated the ability of MIMS to differentiate DNA repair from mitosis². Importantly, all the events included in this manuscript manifested ¹⁵N-thymidine incorporation consistent with mitosis rather than repair.

- We have included a more detailed description of the method. For the changes made in the main text please see page 3 lines 3-19.

1. Fig. 2. The conclusion stated on bottom of page 4 and continuing to top of page 5 that exercised hearts substantially increase cardiomyogenesis seems based on the data showing that there was a trend, lacking statistical significance, towards an increase in the frequency of ¹⁵N labelled, diploid mononucleated cardiomyocytes. These cells are presumed to be cells that have incorporated the thymidine that carried the ¹⁵N label and remained diploid and mononucleated, so hence had not undergone polyploidization or binucleation. I believe that the authors inferred that these cells, by default, must have undergone cell division and for this reason state that there has been an increase in cardiomyogenesis. This is a major claim of the paper. However, if I understand correctly, it is based on a non-significant trend coupled to the absence of evidence for polyploidization or endoreduplication. On balance, this evidence seems rather weak for such a profound conclusion. I think it needs to be either much better explained (perhaps

I am missing something) or better substantiated. Furthermore, the numbers are expressed as percentages rather than the actual number tallied, and it would have been useful to see the actual numbers presented.

We thank the reviewer for the opportunity to clarify our results. In response, we have performed additional experiments which further support our conclusions and we believe considerably strengthen the paper. We agree that ¹⁵N-thymidine labeling is only a measure of DNA synthesis unless advanced assays exclude polyploidization and/or binucleation. We have therefore performed nuclei tracking and ploidy analyses for each of the ¹⁵N-thymidine positive cardiomyocytes in bidirectional serial sections of the ¹⁵N-thymidine positive cardiomyocytes to unambiguously demonstrate the occurrence of newly formed cardiomyocytes. We have now increased the sample size (to a total of n=4 mice per group) and analyzed >1000 cardiomyocytes per group, providing greater statistical power. Consistent with the initial results, we found a similar ratio of ¹⁵N-thymidine labeled/ unlabeled cardiomyocytes in both the additional sedentary and exercised animal. Taken together, the cumulative data are now highly significant statistically (p=0.0003) and demonstrate a substantial increase in ¹⁵N-thymidine-labeled cardiomyocytes in exercised as compared with sedentary hearts (3.62% vs 1.24%). Additionally, in the new studies, we again observed a higher frequency of diploid/mononucleated ¹⁵N-thymidine-labeled cardiomyocytes in hearts from the new exercised compared to the sedentary animal. With an n=4 in each group, we can now report 1.15% diploid/mononucleated ¹⁵N-thymidine-labeled cardiomyocytes in the exercised hearts vs. 0.25% diploid/mononucleated ¹⁵N-thymidine-labeled cardiomyocytes in the sedentary hearts including all the cardiomyocytes analyzed, p=0.01, Fisher's exact test, OR=4.695, CI 1.44 to 15.53), consistent with an exercise-mediated increase in cardiomyogenesis.

In summary, these numbers translate into a 4.6-fold increase in cardiomyogenesis in exercised animals (0.25%/8weeks = 0.0045%/day in sedentary vs. 1.15%/8weeks = 0.021%/day in exercised animals), suggesting that the exercised heart is capable of generating new cardiomyocytes at a projected annual rate of 7.5% vs. 1.63% in sedentary conditions.

- The updated data are now shown in the revised manuscript in new Figure 1b and 2c and the numbers are presented in a table in Figure 1c and 2c.

2. Fig. 3. This figure examines the localization of 15N incorporation relative to the infarct region. Interestingly, an increased proportion of 15N incorporation detected >400 microns from the infarct zone in exercised animal's vs sedentary animals. These data do not seem to have been subjected to the analyses as in Fig. 2, in which cells were discriminated by diploid, mononuclear cells to identify those that are candidates for cells that have undergone cell division. Thus, it seems incorrect to term these cells as having undergone "cardiomyocyte cell cycle activity" (as stated on the first paragraph on page 5). It seems more accurate to characterize these cells as having undergone thymidine

incorporation, which could be due to DNA damage, polyploidization, endoreduplication, etc.

We thank the reviewer for this important feedback. As noted above (please see response to Editor comment #1), MIMS does allow us to distinguish cell cycle events from DNA repair. However, we have now performed additional analyses for ploidy and nucleation of all the ¹⁵N-thymidine positive cardiomyocytes in the extended border zone after myocardial infarction in sedentary and exercised animals to demonstrate a higher ratio of diploid/mononucleated cardiomyocytes, consistent with an increase in cardiomyogenesis.

- The updated data are now shown in the revised manuscript in new Figure 3d and 3e and the numbers are presented in a table in Figure 3f.

3. Fig. 4. The results with LNA to miR-222 are impressive. Interestingly, the authors suggest on top of page 6 that one of the benefits of exercise might be “enhanced repair”. It is puzzling why this is the conclusion here, but not for the experiments in Fig. 3...

We are pleased that the reviewer finds the LNA-miR222 results impressive. We apologize for the lack of clarity regarding the use of the wording: “enhanced repair” on page 6 of our initially submitted manuscript. This wording was used in the context of enhanced cardiomyogenesis and cardiac tissue repair (regeneration) in the extended border region of the infarct as a possible cue to the well-documented benefits of exercise after myocardial infarction.

- For the changes made in the main text please see page 7 lines 7-9 and page 8 lines 5-8.
- The following changes have been made to the Discussion, page 11 line 4:

“Moreover, these studies indicate that miR-222 is necessary for exercise-induced cardiomyogenesis and suggest that enhanced cardiac regeneration may contribute to benefits of exercise after myocardial injury.”

...The difficulty in separating repair from cell division mechanisms is a problem in this paper that needs to be resolved.

Please see our response to this reviewer’s first comment above and also in the response to the Editor’s comment #1.

Reviewer #2

Reviewer #2 Comment

This is an interesting and potentially important manuscript from two leading Harvard groups. This work builds on previous studies by the Rosenzweig lab on the effect of exercise, and the Lee lab using the MIMS approach to determine the rate of cardiomyocyte turnover. The manuscript outlines the cardiomyocyte turnover dynamics following 8 weeks of wheel exercise. The authors demonstrate clearly that there is in fact an increase in cardiomyocyte turnover with exercise. Although this increase in turnover rate is modest, it is an important finding that is worth pursuing.

We thank the reviewer for his/her interest in our study and for the constructive comments to improve our study.

Major issues:

1) The implication of exercise in cardiomyocyte proliferation is intriguing in light of recent reports by the Belmonte group in Zebrafish, and the Sadek group in mice outlining the effect of hypoxia on cardiomyocyte proliferation. As such, the authors need to determine whether exercise activates a hypoxic stress response pathway in cardiomyocytes. Similarly, other critical pathways that have been implicated in cardiomyocyte proliferation such as the Hippo pathway, or Neuregulin need to be explored. As it stands, the mechanistic aspect of the current manuscript is thin.

2) The implication of miR222 is of interest, however the studies as they stand are somewhat superficial. What are the miR222 targets that mediate this observed effect on cardiomyocyte proliferation? does it regulate any of the aforementioned pathways? Importantly, if miR222 is required for the observed effect, does miR222 overexpression result in cardiomyocyte proliferation? This is a critical point in light of a recent report (Su et al, Cell Physiol Biochem 2016;39:1503-1511) which indicates that miR222 overexpression results in heart failure.

To provide additional mechanistic insight and address the questions raised, we have performed further experiments to examine the suggested pathway and their relationship to miR-222 inhibition. In addition, our previous published work implicated p27 and HIPK1 as direct targets of miR-222 that modulate cardiomyocyte proliferation³. To determine whether these targets are involved in our current observations, we also examined their expression in heart samples from the current series of experiments. Our new data demonstrate that after eight weeks of endurance exercise, HIPK1 gene expression remains significantly inhibited, without a significant change in p27 or HIPK2 expression. Moreover, HIPK1 expression increased significantly with miR-222 inhibition (which abolished exercise-induced cardiomyogenesis) while p27 and HIPK2 expression

did not. Taken together with our previously published data demonstrating that HIPK1 is a direct target of miR-222 with anti-proliferative effects in cardiomyocytes³, these data strongly suggest that HIPK1 contributes to miR-222's modulation of exercise-induced cardiomyogenesis.

Our new analyses of other pathways provide additional insight. We investigated a panel of hypoxia-induced genes^{4,5}. In this study, we found a sustained increase in Slc2a (GLUT1) and LDHA expression after eight weeks of exercise, pointing to a significant metabolic adjustment potentially due to increased Hif1a activity. However, these changes were not significantly affected by miR-222 inhibition and thus unlikely to be mediators of miR-222's effects in this context. In contrast, we did find that miR-222 inhibition increased expression of CCNG2 (Cyclin G2), but could not find CCNG2 as a predicted miR-222 target (through Pictar and Targetscan). For this revision, we also specifically checked CCNG2 expression in miR-222 overexpressing neonatal rat ventricular cardiomyocytes (NRVM), and found an increase in CCNG2 expression, further confirming that this is unlikely a direct miR-222 target.

Cell cycle regulators that are reportedly downstream effectors of the Hippo pathway, another pathway implicated in cardiomyocyte proliferation, were increased in exercised hearts. While CDK1, CCNB1 and TEAD2, for example, were all upregulated with exercise, miR-222 inhibition caused further increase of those Hippo/cell cycle regulators. We now specifically ran a pathway analysis from a microarray comparing NRVM with miR-222 overexpression to controls and found cell cycle regulators in general, and with subsequent analysis, TEAD2 specifically, to be upregulated with miR-222 expression, suggesting that they are not direct targets of miR-222's effects in this context. While interesting, further exploration of the role of the Hippo pathway in endurance exercise is beyond the scope of the current study.

In terms of miR-222 overexpression, we have previously described results from miR-222 cardiac transgenic mice, which demonstrated that miR-222 is not sufficient to induce cardiomyogenesis at baseline and did not cause cardiac dysfunction. We are aware of the *Cell Physiol Biochem* paper⁶ but unsure how to interpret their results given that only one transgenic line was reported and thus deleterious effects due to transgene position/ insertion cannot be excluded. We have generated multiple independent miR-222 transgenic lines with results consistent with those reported here. The only instance in which we have observed dysfunction in any of these lines is when we induced miR-222 at birth, leading to a >200-fold increase in miR-222 expression after 3-4 months without evidence of increased cardiomyocyte proliferation. Of note, high-level expression of irrelevant genes (including GFP) has similarly been reported to induce cardiac dysfunction in such transgenic systems⁷. Thus our synthesis of all the available data suggest that miR-222 is not sufficient to induce cardiomyogenesis and does not cause cardiac dysfunction when overexpressed at reasonable levels in the heart.

- These data are now shown in the revised manuscript in new Figure 4 e-f and Supplemental figures 4 a-f are further discussed on page 9-10

3) The findings of the spatial distribution of new cardiomyocytes is of interest. If I understand this correctly, the authors did not observe an increase in cardiomyocyte proliferation in the border zone post MI. If this is the case, this would be contrary to what is known about post-MI induction of cell cycle entry of cardiomyocytes in the border zone. In fact, I believe that the Lee group showed in the original Nature report using the MIMS approach, that turnover in the borderzone is enhanced. This is an important point that needs clarification. If there is no increase in the borderzone, this needs to be explained experimentally.

We thank the reviewer for giving us an opportunity to clarify our findings. The reviewer is correct in pointing out that our group has previously shown, using MIMS approach, that cardiomyocyte ¹⁵N-thymidine incorporation in the peri-infarct region is enhanced. In this study, we confirmed this observation, namely an increase in ¹⁵N-thymidine labeled cardiomyocytes in both sedentary and exercised mice in the peri-infarct region. However, there was no significant difference between the two groups in the number of labeled cardiomyocytes in this area (22.8% vs. 20.4% in sedentary and exercise respectively). We then expanded our search laterally, away from the peri-infarct, into the extended border zone region (>400 μm from the infarct) and here we found that exercise led to a significantly higher proportion of ¹⁵N-thymidine labeled cardiomyocytes (19.1% ¹⁵N-thymidine labeled cardiomyocytes in exercised vs. 5.3% in sedentary mice, p<0.0001). Thus, even though there was no difference in the peri-infarct region between exercised and sedentary hearts, the dramatic increase in cardiomyocyte cell cycle activity seen in the extended region with exercise could explain the favorable influence of cardiac remodeling and likely contributes to the well-documented benefits of exercise in this setting. With this revision, we have also performed the extensive analysis for ploidy and nucleation of all the ¹⁵N-thymidine positive cardiomyocytes in the extended border zone of the infarct and demonstrate an increase of diploid/mononucleated cardiomyocytes consistent with an increased cardiomyogenic response.

- The updated data are now shown in the revised manuscript in new Figure 3d and 3e and the numbers are presented in a table in Figure 3f.

4) As it stands, the conclusions are a bit inflated. For example, given that humans usually do not maintain such a rigorous exercise program for years, it is unclear how impactful an absolute increase in cardiomyocyte proliferation by 2% would achieve. For example, statements in the abstract such as "powerful physiological regulator" and "stimulation of cardiomyocyte proliferation likely contributes to the benefit of exercise" are not supported by the current findings.

We thank the reviewer for giving us an opportunity to clarify our results and conclusions, and we have now modified such statements in the abstract and throughout the paper.

References:

- 1 Gates, K. S. An overview of chemical processes that damage cellular DNA: spontaneous hydrolysis, alkylation, and reactions with radicals. *Chemical research in toxicology* **22**, 1747-1760, doi:10.1021/tx900242k (2009).
- 2 Senyo, S. E. *et al.* Mammalian heart renewal by pre-existing cardiomyocytes. *Nature* **493**, 433-436, doi:10.1038/nature11682 (2013).
- 3 Liu, X. *et al.* miR-222 is necessary for exercise-induced cardiac growth and protects against pathological cardiac remodeling. *Cell Metab* **21**, 584-595, doi:10.1016/j.cmet.2015.02.014 (2015).
- 4 Bohuslavova, R. *et al.* Gene expression profiling of sex differences in HIF1-dependent adaptive cardiac responses to chronic hypoxia. *J Appl Physiol (1985)* **109**, 1195-1202, doi:10.1152/jappphysiol.00366.2010 (2010).
- 5 Semenza, G. L. Targeting HIF-1 for cancer therapy. *Nature reviews. Cancer* **3**, 721-732, doi:10.1038/nrc1187 (2003).
- 6 Su, M. *et al.* Cardiac-Specific Overexpression of miR-222 Induces Heart Failure and Inhibits Autophagy in Mice. *Cell Physiol Biochem* **39**, 1503-1511, doi:10.1159/000447853 (2016).
- 7 Huang, W. Y., Aramburu, J., Douglas, P. S. & Izumo, S. Transgenic expression of green fluorescence protein can cause dilated cardiomyopathy. *Nature medicine* **6**, 482-483, doi:10.1038/74914 (2000).

Reviewer #1:

Remarks to the Author:

NCOMMS-17-19078A

The revised version satisfies my concerns.

Reviewer #2:

Remarks to the Author:

The authors have now addressed most of my concerns. Several issues remain:

1) The authors did not sufficiently modify their statements in the abstract and discussion. The exercise program used does not in any way reflect a real life exercise scenario and as such it is difficult to understand how these result can be extrapolated to an exercise program that humans use. It is understandable that this is a catchy statement, however 5 km/day for a mouse is far more than an exercise program that even endurance athletes are exposed to. Therefore I still recommend tempering the general statement about exercise making new cardiomyocytes.

2) The effect of miR 222 is not very convincing as the primary mechanism. It is true that it modulates the effect of exercise, but this just proves that it is a cell cycle regulator, and not necessarily THE modulator of the exercise-induced response. The authors should highlight in their discussion that based on their own results (for example the hypoxia target genes result, and CDK1 results) that the effects on cell cycle are not limited to miR222

The authors have now addressed most of my concerns. Several issues remain:

1) The authors did not sufficiently modify their statements in the abstract and discussion. The exercise program used does not in any way reflect a real life exercise scenario and as such it is difficult to understand how these result can be extrapolated to an exercise program that humans use. It is understandable that this is a catchy statement, however 5 km/day for a mouse is far more than an exercise program that even endurance athletes are exposed to. Therefore I still recommend tempering the general statement about exercise making new cardiomyocytes.

We thank the reviewer for his/her comments. We further tempered our statement and discussed that the voluntary wheel running exercise program that our mice underwent may be difficult to compare to human exercise programs.

Please see Abstract page 2 line 13-14 and Discussion page 12 line 15-16.

2) The effect of miR 222 is not very convincing as the primary mechanism. It is true that it modulates the effect of exercise, but this just proves that it is a cell cycle regulator, and not necessarily THE modulator of the exercise-induced response. The authors should highlight in their discussion that based on their own results (for example the hypoxia target genes result, and CDK1 results) that the effects on cell cycle are not limited to miR222

We have now highlighted more clearly that we describe one seemingly important mechanism for cardiomyogenesis, miR222, but that our data suggests that other regulators may play important roles as well.

Please see page 12 line 5-6.